# Environmental monitoring and health assessment in an industrial town in central India: A cross-sectional study protocol

**Tanwi Trushna**[1]*, **Vikas Dhiman**[1], **Satish Bhagwatrao Aher**[2], **Dharma Raj**[3], **Rajesh Ahirwar**[4], **Swasti Shubham**[5], **Subroto Shambhu Nandi**[2]*, **Rajnarayan R. Tiwari**[6]

**1** Department of Environmental Health and Epidemiology, ICMR-National Institute for Research in Environmental Health, Bhopal, Madhya Pradesh, India, **2** Department of Environmental Monitoring and Exposure Assessment (Air), ICMR-National Institute for Research in Environmental Health, Bhopal, Madhya Pradesh, India, **3** Department of Environmental Biostatistics and Bioinformatics, ICMR-National Institute for Research in Environmental Health, Bhopal, Madhya Pradesh, India, **4** Department of Environmental Biochemistry, ICMR-National Institute for Research in Environmental Health, Bhopal, Madhya Pradesh, India, **5** Department of Environmental Pathology, ICMR-National Institute for Research in Environmental Health, Bhopal, Madhya Pradesh, India, **6** ICMR-National Institute for Research in Environmental Health, Bhopal, Madhya Pradesh, India

* s.nandi76@icmr.gov.in (SSN); tanwitrushna@gmail.com (TT)

**Data Availability Statement:** No datasets were generated or analysed during the current study. All relevant data from this study will be made available upon study completion.

## Abstract

### Background

Textile industry has been widely implicated in environmental pollution. The health effects of residing near manufacturing industries are not well documented in India, especially in central India. Hence, a cross-sectional environmental monitoring and health assessment study was initiated as per directions of the local authorities.

### Methods

Comprehensive exposure data about the concentrations of relevant pollutants in the ambient air and ground water samples in the study area will be collected over one year. Using stratified random sampling, 3003 apparently healthy adults will be selected from the study area. Sociodemographic and anthropometric information, relevant medical and family history, and investigations including spirometry, electrocardiogram, neurobehavioral tests, and laboratory investigations (complete blood count, lipid profile and random blood glucose) will be conducted. Finally Iodine azide test and heavy metal level detection in urine and blood samples respectively will be conducted in a subset of selected participants to assess individual pollution exposure. Ethics approval has been obtained from the Institutional Ethics Committee of the National Institute for Research in Environmental Health (No: NIREH/IEC-7-II/1027, dated 07/01/2021).

### Discussion

This manuscript describes the protocol for a multi-disciplinary study that aims to conduct environmental monitoring and health assessment in residential areas near viscose rayon

**Funding:** The study on which this protocol is based was funded as a consultancy project by the Aditya Birla Grasim Industry Ltd (Service No. 4700221679/104; dated 10.11.2020) [Funding issued to Subroto Shambhu Nandi] which owns the major viscose rayon manufacturing unit in the study area (Birlagram Nagda) as per the directions of local pollution control authorities. The funders had and will not have a role in study design, data collection and analysis, decision to publish, or preparation of the manuscript.

**Competing interests:** I have read the journal's policy and the authors of this manuscript have the following competing interests: The study on which this protocol is based was funded by a commercial organisation (Aditya Birla Grasim Industry Ltd) which owns the major viscose rayon manufacturing unit in the study area (Birlagram Nagda) as per the directions of local pollution control authorities. However, the funding agency has no role in design/conduct of the study, decision to publish, or preparation/submission of this manuscript and this does not alter our adherence to PLOS ONE policies on sharing data and materials. No author received any direct payment from the industry with regards to their contribution to this manuscript or the study as a whole, in terms of employment or individual consultancy charges. The authors have no other relevant declarations relating to patents, products in development, marketed products, etc.

**Abbreviations:** Al, Aluminium; ANOVA, Analysis Of Variance (Statistical Test); BMI, Body Mass Index; Cl, Chloride Ion; Cl2, Chlorine Gas; CPCB, Central Pollution Control Board; $CS_2$, Carbon Disulphide; ECG, Electrocardiogram; $FEV_1$, Forced Expiratory Volume In first Second; FVC, Forced Vital Capacity; $H_2S$, Hydrogen Disulphide; HCl, Hydrogen Chloride; Hg, Mercury; HNO3, nitric acid; MP, Madhya Pradesh; Pb, Lead; PFT, Pulmonary Function Test; pH, Potential Of Hydrogen (Measure Of Acidity/Basicity); ppb, parts per billion; $SO_2$, Sulphur Dioxide; $SO_3$, Sulphur Trioxide; SO42, Sulphate Ion; °C, Degrees Celsius; km, Kilometre; kg/m2, Kilogram Per Square Metre; mL, Millilitre; $mgL^{-1}$, Milligram Per Litre; mm Hg, Millimetre Of Mercury.

and associated chemical manufacturing industries. Although India is the second largest manufacturer of rayon, next only to China, and viscose rayon manufacturing has been documented to be a source of multiple toxic pollutants, there is a lack of comprehensive information about the health effects of residing near such manufacturing units in India. Therefore implementing this study protocol will aid in filling in this knowledge gap.

## Introduction

Industrial development has contributed to economic growth but at the same time industrial activities constitute a major source of environmental pollution [1]. Adverse health effects posed by industrial pollution is well-established [1]. De Sario et al., 2018 [2] in their scoping review identified 762 epidemiological studies which conclude that living in industrial areas is associated with a wide range of diseases ranging from malignancies and respiratory diseases to adverse birth outcomes and even premature mortality. Of these studies, few were conducted in India that focussed on the public health and environmental effects of industrial emission. Chatham-Stephens et al., 2013 [3] using the Toxic Sites Identification Program developed by the Blacksmith Institute and the United Nations Industrial Development Organization identified 221 point-sources of pollution from industrial activities in India that pose risk to public health. The authors [3] calculated that exposure to pollutants generated in industrial regions in developing countries like India, Indonesia, and the Philippines pose equal or even greater risk to human health than widespread communicable diseases like malaria and account for almost eight million years of healthy life lost due to disease, disability or premature death (i.e. 828,722 Disability-adjusted Life Years) among the 8.6 million exposed population in the three countries. Therefore, research is needed to assess the environmental health effects of major industries in India.

India, being a rapidly developing economy, is home to multiple industries. Textile industry, plays a critical role in Indian economy as it is the second largest source of employment for a country which is counted amongst the top producers, exporters as well as consumers of textiles in the world [4]. However, the textile industry has been widely implicated in pollution of various environmental matrices like air, water and soil both in India [5, 6] and other countries [7, 8]. It accounts for almost twenty percent of global industrial activity induced water pollution [7] and is associated with emission of many gaseous pollutants into ambient air [9]. Multiple studies [10. 11] have documented the adverse health effects of exposure to the chemical pollutants that emanate from textile and associated chemical manufacturing. In India, research [12–14] on the health effects of such textile industry has mostly been focussed on individuals who are occupationally exposed to the chemical pollutants. The health effects of residing near manufacturing industries are not as well documented.

The industrial area of Birlagram Nagda in the central Indian state of Madhya Pradesh (MP), has been under scrutiny due to the pollution emanating from the multiple viscose rayon textile and associated chemical manufacturing industries operating in the area [15]. As per data compiled by the Central Pollution Control Board (CPCB) [16] in 2013, the apex government run agency monitoring environmental pollutant concentrations in the country, the area has a comprehensive environment pollution index score of 66.67, which indicates that it is among the "severely polluted" industrial clusters in India. Although recent statistics have shown improvement in the area's pollution index, it is still listed in the top hundred polluted industrial clusters in India [17]. Major chemical pollution associated with viscose rayon textile manufacturing occur secondary to gaseous emission of carbon disulphide ($CS_2$), Hydrogen

Disulphide (H$_2$S), Hydrogen Chloride (HCl), Chlorine (Cl$_2$), Sulphur oxides (SO$_2$ and SO$_3$) and untreated discharge of viscose fibre wastewater effluent containing high concentration of heavy metals, and other toxic substances into natural water sources [9]. However, there is hardly any scientific evidence regarding the health effects of residing near the industries of Birlagram. Considering the growing health concerns of the local public, regulatory authorities in India directed the industries operational in the region to implement pollution control strategies as well as commissioned detailed environmental and health-oriented investigation of the issue [18]. The current study was initiated as per directions of the local authorities to evaluate the health effects of exposure to multiple pollutants emitted from the industrial activity in Birlagram. Specifically, the objectives of the current study include:

i. To measure the concentration of gaseous pollutants (i.e. CS$_2$, H$_2$S, HCl, Cl$_2$, and SO$_2$) in the ambient air in the study area.

ii. To measure the concentrations of heavy metals and anions [i.e. lead (Pb), mercury(Hg), aluminium (Al), chloride (Cl$^-$) and sulphate(SO$_4$$^{2-}$)] in groundwater samples.

iii. To determine the association between the concentration levels of pollutants in air and water samples with that in bio-samples (blood and urine).

iv. To identify the association between exposure to pollutants and health effects in study participants.

## Methods and analysis

### Study design

This is the protocol for a cross-sectional research study that aims to collect and correlate comprehensive exposure and health outcome information to ascertain objective achievement (see Fig 1). This study will be conducted during a duration of one year. The time schedule of various study methods of this protocol is described in Fig 2.

### Setting

This study will be conducted in and around the Birlagram industrial region located in Nagda city in the Ujjain district (administrative unit) in the central Indian state of MP. Almost 72 million individuals reside in MP, most (72%) of whom reside in rural areas and depend on agriculture for their livelihood [19]. However, industrial activity in MP is gradually increasing [20] and multiple large scale industries in the Ujjain district [21] constitute an emerging public health concern for the province that needs to be evaluated.

The Birlagram industrial area is situated within the borders of Nagda city which in turn is surrounded on all sides by villages [15]. The industries situated in Birlagram are directly or indirectly (through production of chemical raw materials) involved in synthetic textile (viscose rayon) manufacturing [22]. Including the population residing in Birlagram itself, Nagda city has a total population of more than 0.1 million. River Chambal is the main source of water for the region and previous research findings highlight the poor quality of its water [23]. Details on air quality of the area are lacking in published literature.

### Participant selection and sampling strategy

**Sample size calculation.** As the health effects of residing near textile manufacturing industries in Birlagram, India are not previously documented, we calculated the sample size using the statistics reported by a previous study conducted in Taiwan where authors [24]

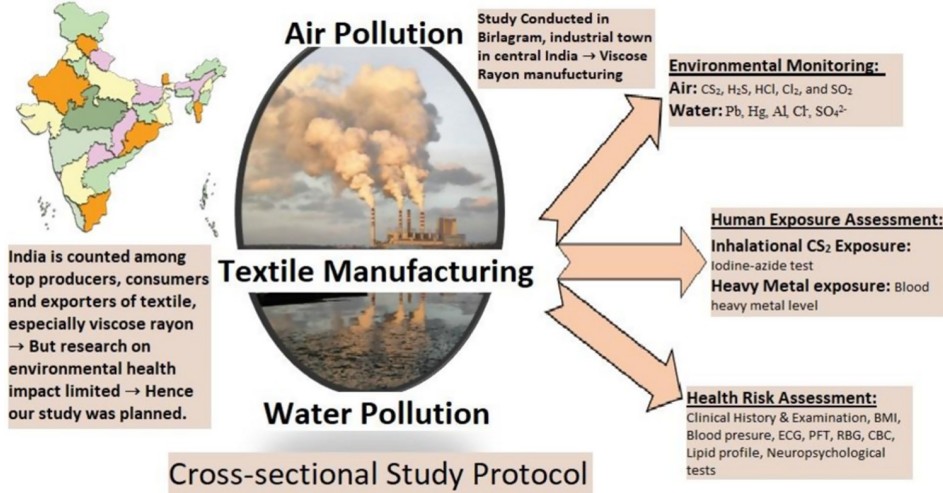

**Fig 1. Graphical abstract: Overview of the study methodology.** [Note: carbon disulphide ($CS_2$), Hydrogen sulphide ($H_2S$), Hydrogen Chloride (HCl), Chlorine ($Cl_2$), Sulphur oxides ($SO_2$), lead (Pb), mercury(Hg), aluminium (Al), chloride ($Cl^-$), sulphate ($SO_4^{2-}$), body mass index (BMI), Electrocardiogram (ECG), Pulmonary function test (PFT), Random Blood glucose (RBG), Complete blood count (CBC)] (The graphical abstract image has been created by the research team using copyright-free images of Indian map and industry from https://pixabay.com).

found that 26.3% of the population exposed to emissions from a viscose rayon manufacturing industry had electrocardiogram (ECG) abnormalities. Since one of our outcomes is the identification of the proportion of exposed population in Birlagram, India with ECG abnormalities, we calculated sample size as follows:

$$\text{Sample size (n)} = \frac{z^2 pq * (1 + R) * \text{Deff}}{d^2}$$

Where, p = Expected population proportion = 0.26; q = 1-p = 0.74; z = z value for 5% level of significance is 1.96; R = Non-response rate (assumed to be 30% as pooled non-response rate

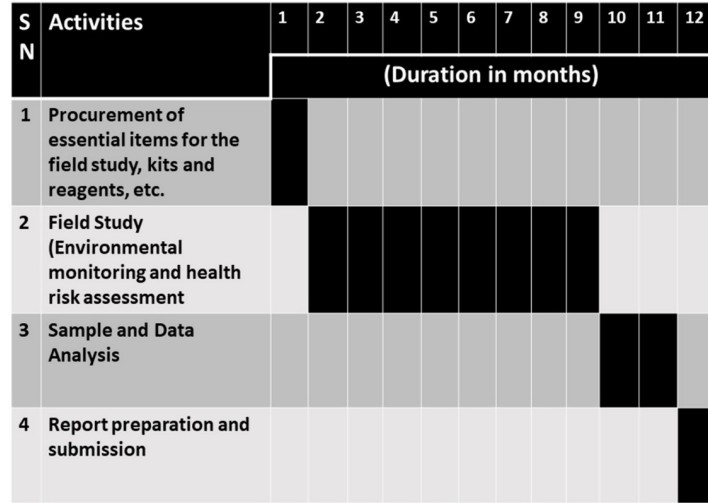

**Fig 2. Time schedule of activities.** [Note: Each numbered column represents one month of the project duration with the total duration of the project being 12 months].

reported by a systematic review of participation response in Indian industrial surveys [25]); Deff = Design effect(1.25 for systematic sampling); and d = Margin of error 2% (assumed to be 0.02). The calculated sample size was 3003.

**Sampling strategy.** In this study stratified random sampling technique will be used for selection of participants where stratification will be done based on environmental factors that are likely to affect exposure. Since both distance from fixed contamination source as well as local environment features such as meteorology, topography, etc. are the major factors affecting the dispersion of both ambient air as well as water pollutants [26], a map of the study area was created to assist in selection of exposure monitoring locations and in participant sampling (see Fig 3). Briefly, taking Birlagram industrial region as the centre, villages located within consecutive circles of radius 2 kilometres (km), 5km and 10km were demarcated on the map using ArcInfo (version 10). This strategy was used as it is expected that gaseous pollutants' dispersion occurs homogenously in all directions independent of anthropogenic roadways and that with increase in distance the pollutant concentration will become diluted enough to prevent adverse health effects [27]. Based on seasonal wind pattern data retrieved from local authorities, the

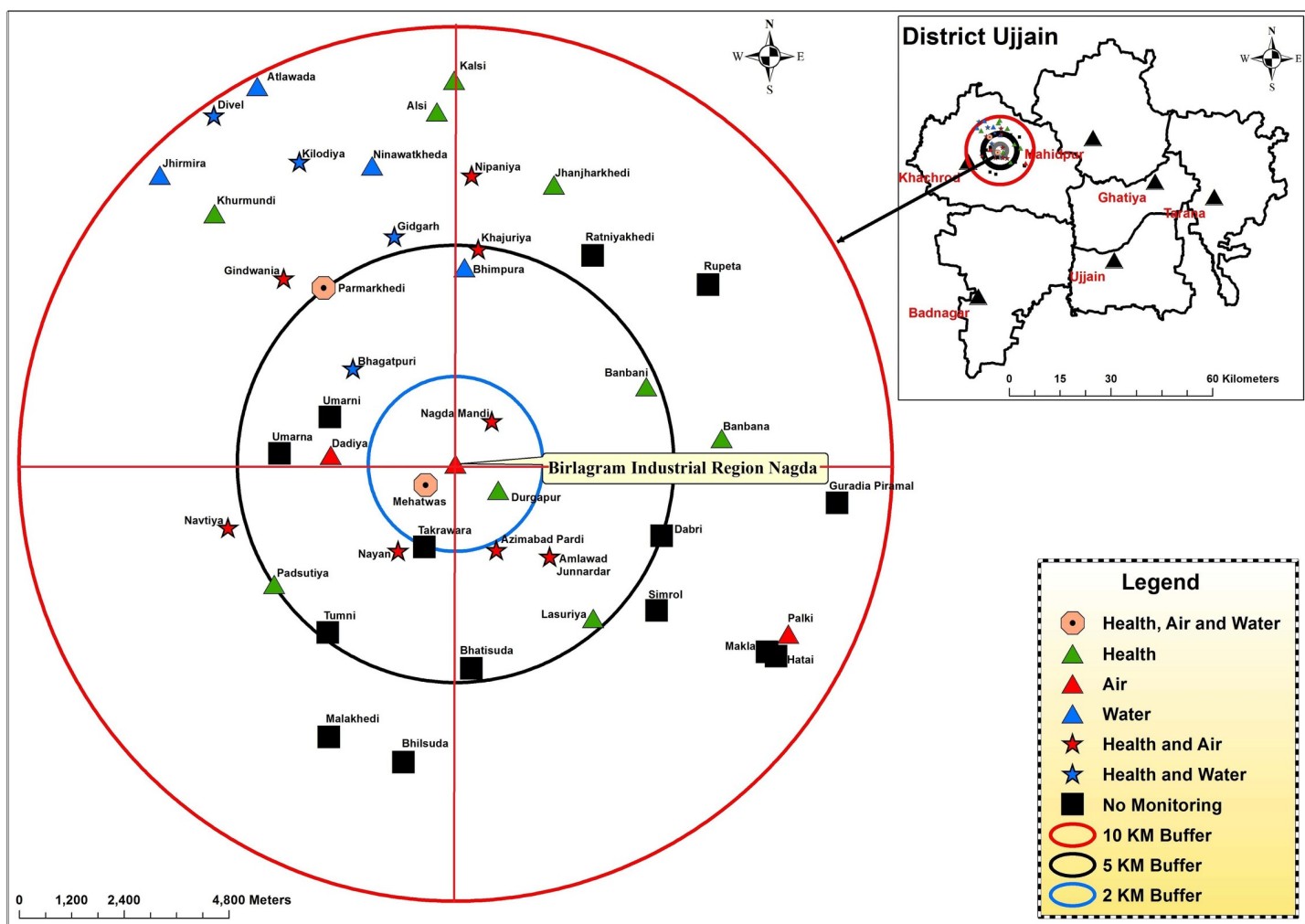

**Fig 3. Map of the study area created by research team to assist in selection of exposure monitoring locations and in participant sampling.** [Note: The inset in the map shows the Ujjain district of the central Indian province of MP. Within the Khachrod Tehsil of the Ujjain district, lies the study area of the current project. Considering the Birlagram industrial region as the centre, three concentric circular regions were demarcated based on distance (2, 5 and 10 km from the centre)].

north-eastern quadrant of the circular study area was seen to be the downwind area for most of the year. The downstream direction of River Chambal falls in the north-western quadrant of the circular study area. Therefore the majority of the pollutants are expected to disperse in the northern direction increasing the potential risk faced by the residents and so, we decided to over-represent the northern hemisphere.

Finally, 2000 participants will be selected from the villages in the northern hemisphere while 1003 will be taken from the villages in the southern hemisphere within the 10km radius comprising a total of 3003 participants. In each hemisphere, a fixed number of villages will be selected randomly from those situated in each of the 0-2km, 2-5km and 5-10km zones. Further in each selected village, using the list of all individuals residing in the villages that will be retrieved from the local authorities as the sampling frame, individuals meeting the eligibility criteria will be randomly invited for participation in the study. Total number of individuals sampled in each village will be proportionate to the total population of the selected village.

Participants will be selected for the study only if they fulfill all the following criteria:

- Adults ($\geq$18 years of age) of both genders (except pregnant women),

- Should have been residing for at least one year in the selected village,

- No history of neuropsychiatric conditions that might hinder completion of neurobehavioral tests,

- No history of other conditions that are contraindications for performing Pulmonary function test (PFT) like presence of any active disease (e.g. pulmonary Tuberculosis, active haemoptysis, orofacial pain, acute respiratory infection including COVID-19 i.e. coronavirus disease caused by SARS-Cov-2 virus) or recent history (within previous 1 month) of myocardial infarction/aneurysm/surgery, etc.

**Environmental monitoring of pollutants at study sites.** Pollutants for assessment in both environmental and human matrices were selected based upon their relevance to viscose rayon and associated chemical manufacturing industry identified through review of previously published literature [9].

*i) Monitoring of gaseous pollutants in ambient air*. The ambient air monitoring will be carried out in the study area (see Fig 2) for identified gaseous pollutants viz., $CS_2$, $H_2S$, $HCl$, $Cl_2$, and $SO_2$ using a factory calibrated, electrochemical sensor based multi-gas monitor (Make: Swan® Environmental Pvt. Ltd. India, Model: GRI-IAT; Sensor make: Membrapor®, Switzerland). The monitoring will be carried out in post-monsoon season (October-January). The instrument has an inbuilt geographical positioning system to record the coordinates of sampling site. The ambient air monitoring will be carried out continuously for 24 hours in at identified sites. In case of cluster of sites, a representative site will be identified for gaseous monitoring. The monitor will be placed at more than 3 metres height from ground level at a place having uninterrupted power supply. The data loggers present in the instrument will store the data at set frequency and will be utilized for further statistical analysis. The sampling and monitoring of ambient air for said pollutants will be carried out following the existing guidelines of CPCB, Ministry of Environment, Forests and Climate Change, New Delhi, India [28].

*ii) Monitoring of pollutants in groundwater*. Levels of concerning heavy metals and ions, namely Al, Pb, Hg, $Cl^-$ and $SO_4^{2-}$ will be assessed in representative groundwater samples collected from sources including tube-wells, hand-pumps and open wells from the selected 10 downstream villages (Fig 2). 120 millilitre (mL) each of groundwater samples will be collected from all available and accessible sources in each village. Samples will be collected in high

density polypropylene (HDPE) bottles which were pre-cleaned by soaking overnight in 2% nitric acid and washed with double distilled water. Water samples will be collected from hand-pumps, tube-wells, and open-wells after removing stagnant water using appropriate purging methods [29]. Also, a few water samples will be collected from the Chambal River at specified locations to assess metallic pollutants. Collected water samples will be labeled and transported to the laboratory at room temperature for initial measurement of physiological parameters including pH, total dissolved salts, conductivity, and salinity. Subsequently, the samples will be preserved with trace metal grade nitric acid ($HNO_3$, pH <2) and stored at 4°C till analysis within a week.

Analysis of toxic metals, viz. Pb, Hg and Al in the collected groundwater samples will be carried out using inductively coupled plasma optical emission spectroscopy (iCAP® 7400 Duo ICP-OES, ThermoFisher Scientific® Pvt Ltd) using the EPA method 200.7 (revision 4.4) [30]. Briefly, 6-point calibration standards for each element (Al, Pb, and Hg) will be first prepared in the range of 1–200 parts per billion (ppb) by serially diluting the multi-element stock solution (ICP multi-element standard XIII, Merck, Germany) in 5% $HNO_3$ in double distilled water. Specific run parameters and sample introduction system that will be used for analysis of these metals on ICP-OES instrument are summarized in Table 1. The concentration of individual metals in the unfiltered acid preserved groundwater samples (turbidity <1 Nephelometric Turbidity unit), will be determined after successful run of standards (e.g., linear graph with correlation coefficient > 0.999) in the axial view using standard sample introduction setup for Al and Pb, and hydride generation sample introduction system for Hg. The online hydride generation for Hg will be achieved with Enhanced Vapour System sample introduction kit using 0.5% mass by volume (m/v) sodium tetrahydroborate ($NaBH_4$) stabilized in 0.5% m/v Sodium hydroxide (NaOH) and 50% volume/volume (v/v) HCl solution [31]. Emission acquisition will be done on the Qtegra® ISDS Software at interference free wavelengths of 220.353 nanometer (nm), 184.950 nm, and 396.152 nm for Pb, Hg and Al, respectively. The instrument performance will be checked in between the sample run using quality control samples containing metals of known concentration. Metal levels in each sample will be reported in ppb levels.

**Table 1. The ICP OES instrument parameters that will be used during analysis of Pb, Al, and Hg in the groundwater and/or blood samples.**

| Parameter | Aqueous sample introduction | EVS sample introduction |
|---|---|---|
| Pump Tubing | Sample = white/white | Sample = green/green |
| | Drain = orange/blue | Drain = black/white |
| | | Acid blank = orange/yellow |
| | | Reductant = black/black |
| Spray Chamber | Baffled cyclonic | Gas Liquid Separator |
| Nebulizer | V-groove | V-groove |
| Auxiliary Gas Flow | 0.5 Litre per minute | 0.5 Litre per minute |
| Coolant Gas Flow | 12 Litre per minute | 16 Litre per minute |
| Nebulizer Gas Flow | 0.5 Litre per minute | 0.3 Litre per minute |
| Pump Speed | 50 revolutions per minute | 30 revolutions per minute |
| Center Tube | 1.5 millimeter | 1.5 millimeter |
| RF Power | 1150 Watt | 1350 Watt |
| Exposure Time | Axial view: Ultraviolet light- 15 seconds, Visible light- 15 seconds | Axial view: Ultraviolet light- 15 seconds, Visible light- 15 seconds |
| | Radial view: Ultraviolet light- 15 seconds, Visible light- 15 seconds | Radial View: Ultraviolet light- 15 seconds, Visible light- 15 seconds |

EVS: Enhanced Vapour System sample introduction, Acid blank: 50% v/v HCl; Reductant: 0.5% m/v $NaBH_4$ in 0.5% m/v NaOH. Aqueous sample introduction kit was used for analysis of Al and Pb. Hydride generation kit (EVS) was used for analysis of Hg.

Analysis of chloride and sulphate ions will be performed through kit-based colorimetric (AQUACheck chloride testing kit, WT004A, Himedia Laboratories Pvt Ltd, India) and turbidimetric methods (AQUACheck sulphate testing kit, WT043, Himedia Laboratories Pvt Ltd, India), respectively [32, 33]. As per manufacturers' instruction, detection of sulphate ions will be performed by taking 1 mL of water sample in a test jar to which 5 drops of supplied reagent 043A and one spoonful of reagent 043B will be added. After a gentle mixing, the solution will be incubated for 5–10 minutes at room temperature to allow the reaction to occur. Then, the solution will be diluted to 10 mL and the test jar will be placed on a circular control sulphate disk of colour chart to observe black disk from above and match with the standard sulphate disk to find out sulphate levels as milligram per litre or parts per million by turbidity comparison. Similarly, for detection of chloride, 2 mL of water sample will be taken to test jar and added with spoonful of reagent 04A-1 and two drops of reagent 04A-2. After gentle mixing, reagent 04A-3 will be added drop by drop to this solution until the colour of entire solution will changes to bluish violet. Chloride content of the water sample will then be calculated using the formula:

$$Cl^-(ppm) = 50 \times Number\ of\ drops\ of\ 04A-3\ solution$$

**Human exposure assessment.** Following biomonitoring will be done to assess exposure in the selected participants:

*i) Inhalational $CS_2$ exposure.* Iodine-azide test will be carried out for estimation of $CS_2$ exposure of the participants. The iodine-azide test is based on the fact that certain constituents in the urine of persons exposed to $CS_2$ catalyze the reaction between iodine and sodium azide [34, 35]. Participants will be asked to provide approximately 50 mL of clean catch spot urine sample in a sterile container which will be stored at -20˚C till analysis. Preparation of reagents and buffers as well as the test procedures will be carried out as previously described [34, 36]. The exposure coefficient will be calculated to estimate Carbon disulfide exposure.

*ii) Metal (Pb, Hg and Al) exposure.* Blood levels of Pb, Al, and Hg will be examined in a subgroup of the studied population (sample from every tenth participants). For this, 1 mL of whole blood from each participant will be mixed with 6 mL of freshly prepared mixture of concentrated trace metal grade $HNO_3$ and hydrogen peroxide ($H_2O_2$) in a ratio 2:1 (v/v) in high-purity polytetrafluoroethylene (PTFE-TFM) vessels. After gentle mixing of reactants, the PTFE-TFM vessels will be arranged in the rotor (24HVT80, Anton PAAR) and digestion will be carried out in the Anton Paar, Multimicrowave PRO Reaction System at 200˚C for 15 minutes [37]. After digestion, the solutions resulting from each digested samples will be cooled to 40˚C and diluted to 30 mL with distilled water. Blank will also be prepared for each cycle of digestion using distilled water, nitric acid, and hydrogen peroxide mixture. Analysis of Pb, Al and Hg in the digested blood samples will be carried out on ICP-OES (iCAP® 7400 Duo ICP-OES, ThermoFisher Scientific® Pvt Ltd) system using same run parameters as used for groundwater analysis. Recorded metal levels in each sample will be tabulated in ppb levels.

**Health assessment.** Following health-related data will be collected from all selected participants:

*i) Questionnaire-based collection of data on demography, exposure and lifestyle.* A structured questionnaire which has been developed by the researchers will be pilot-tested and thereafter wards used for collection of socioeconomic and demographic details along with information regarding lifestyle factors, occupational and relevant exposure history. The questionnaire will be administered by trained investigators in the local language.

*ii) Anthropometry and blood pressure monitoring.* Height and weight will be measured using stadiometer and weighing scale, respectively and will be used to calculate body mass

index (BMI). As per guidelines for Indian population [38], BMI in the range of 18.5–22.9 kilogram per square metre ($kg/m^2$) will be considered as normal, 23–24.9 $kg/m^2$ as overweight and more than or equal to 25 $kg/m^2$ will be considered as obesity. Blood pressure will be measured as per the International Society of Hypertension's 2020 Global Hypertension Practice Guidelines [39] in sitting position on the left arm using validated electronic blood pressure monitor (Omron®), after 5 minutes of rest. Systolic and diastolic blood pressure will be recorded as the mean of last two of three readings taken 1 minute apart. Hypertension will be defined as the study participants having raised systolic and/or diastolic blood pressure (more than or equal to 140 and/or 90 millimetres of mercury i.e. mm Hg, respectively) [38].

*iii) Clinical history and examination.* The information regarding current and previous cardiorespiratory and neurological symptoms will be taken by the physicians from the participants using a clinical proforma. The details regarding the onset, duration, severity of symptoms, diurnal variation, aggravating/relieving factors, and associated symptoms details will be noted. A general physical examination and detailed respiratory, cardiological and neurological examination of all participants will be conducted which will include higher mental function examination, speech assessment, cranial nerve examinations, examination of the motor system and muscle power, examination of the sensory system, reflexes and coordination, and gait assessment, etc. All abnormal findings will be recorded in a pre-designed the clinical proforma.

*iv) Neurobehavioral tests.* $CS_2$, mainly used in the production of viscose rayon fibre, is one of the main air pollutant in Nagda Industrial town. Although overt $CS_2$-induced neurotoxicity is no longer a serious problem nowadays, neurobehavioral problems due to chronic exposure to low levels of $CS_2$ are not uncommon [40]. Previous studies, although mostly conducted in occupational set-up, have shown problems in cognition [41, 42], fine coordination [43], learning and memory [44], visual attention & concentration [45], sensory and motor functions [46], and other behavioral functions [47] in humans due to $CS_2$ exposure. Hence a battery of neurobehavioral tests (see Table 2) will be employed in the present study to screen any known neuro-behavioral effects due to $CS_2$ exposure.

*v) Clinical investigations (PFT and ECG).* PFT will be conducted using portable spirometer (Cosmed® Pony FX) on comfortably seated participants by a trained technician, supervised by a trained physician, following national guidelines [56], to measure forced expiratory volume in first second ($FEV_1$), forced vital capacity(FVC), and peak expiratory flow. Of three acceptable spirograms defined as the difference between the two largest FVC and the two largest $FEV_1$ measurements are less than or equal to 0.150 litres, the best value for each parameter will be included for analysis. The predicted values for the FVC will be calculated for each subject using Kamat's Regression equation [57]. For categorizing pulmonary function impairment, cut-off of 80% of predictive FVC and 70% of $FEV_1$/FVC ratio will be taken [58].

A portable 12-lead Electrocardiogram device (BPL CardiArt® 9108D) will be used to measure electrical activity of the heart following American Heart Association guidelines [59]. Briefly, in a private screened area, ECG recording will be done by previously trained medical staff on the chest and limbs of supine and relaxed participants. Trained physicians will interpret the ECG recording as per standard guidelines [60].

*vi) Laboratory investigations.* A battery of clinical laboratory investigations comprising of Random Blood Glucose, Complete Blood Counts, and Lipid profile (Total cholesterol, High and low density lipoprotein cholesterol, and Triglyceride) will be carried out for all participants of the study. Blood sample collection will be carried out by trained technician according to standard operating procedure in compliance with WHO Best Practices [61].

To undertake the afore-mentioned three tests, briefly, a total of 5 mL sample of venous blood will be collected from the median cubital vein of the participants by trained technician

**Table 2. Details of neurobehavioral tests to be administered to study participants.**

| Name of the test | Domain measured | Method/instructions | Interpretation |
|---|---|---|---|
| Hindi Mental State Examination (HMSE) | Cognition | This is a validated 23 items questionnaire and widely used for assessing the cognitive impairment in Indian population [48]. | Sum total of the correct responses. |
| Finger dexterity test | Fine coordination | We will adopt the method used by Tiwari R. R. et. al. (Tiwari RR, Raghvan S, Tripathi S. Cardiological and neurological health effects in viscose rayon workers exposed to carbon disulphide. 2012. [Unpublished]) to perform finger dexterity test. The subject is given the instructions to pick three pins at a time with his/her preferred hand and go on filling into the holes row wise as fast as he/she can do in 180 seconds. | The score is the number of holes filled in 180 seconds. The higher the score, the more is the efficiency in performance. |
| Tweezer dexterity test | Fine coordination using a tweezer | We will adopt the method used by Tiwari R. R. et. al. (Tiwari RR, Raghvan S, Tripathi S. Cardiological and neurological health effects in viscose rayon workers exposed to carbon disulphide. 2012. [Unpublished]) to perform tweezer dexterity test. The subject is given the instructions to pick up with the tweezer one pin at a time with his/her preferred hand and begin to fill into the holes row wise as fast as he/she can do in 180 seconds. | The score is the number of holes filled in 180 seconds. The higher the score, the more is the efficiency in performance. |
| Letter-digit substitution test (LDST) | Verbal learning | The LDST consists of a worksheet, which has 8 rows and 20 columns and randomly arranged letters in rows and columns. We will follow previously published method to perform LDST. The subject will be first given the instructions to substitute target letters with digits and practice on a separate sheet for 1–2 rows. Then he/she will be instructed to substitute as many letters with digits as possible in 180 seconds [49]. | The net score will be obtained by deducting wrong substitutions from the total substitutions attempted [50]. |
| Memory forward and backward test | Forward span: attention efficiency and capacity. Backward span: executive task particularly dependent on working memory. | We will follow previously published methodology [51] to perform this test. A list of random numbers is read aloud and the subject is asked to recall the numbers in the same order (forwards) and then in reverse order (backwards). | The score will be reported as the maximum number of digits correctly produced forwards (forward subscore) and backwards (backward subscore) by the subject. |
| Serial three subtraction test | Concentration and attention | We will follow previously published methodology [52] to perform this test. The subject will be asked to count the numbers by serial subtractions of three from 20 backwards. | The time to complete the task along with errors will be noted as score [53]. |
| Trail making test (TMT) | Trail making test A: visual attention and processing speed Trail making test B: executive abilities including set shifting and mental flexibility | We will follow previously published method [54] to perform TMT. In Trail A, the subjects are asked to draw lines connecting circled numbers on a page in sequence as quickly as they can. In Trail B, one alternates between numbers and letters. | In each test, the primary score is the time to complete the task. |
| Finger tapping test (FTT) | Motor functioning, specifically, motor speed and lateralized coordination | We will use a shorter version of Finger tapping test [55]. In this test, the subject's palm of dominant hand should be immobile and flat on the board, with fingers extended, and the index finder placed on the counting device. The subject taps his index finger on the lever as quickly as possible within a 10 seconds time interval, in order to increase the number on the counting device with each tap. Such five trials are performed with 10 seconds rest interval in-between each trial [55]. | The score is the mean score of trials 3–5. |

maintaining sterile precautions. 2 mL blood will be collected in ethylene diamine tetra acetic acid vacutainer for complete blood count which will be processed within 4 hours using a fully automated hematology analyzer (Sysmex®). 3 mL blood sample will be collected in plain

vacutainer from which serum will be separated and stored at -20°C till analysis. The serum will be used for estimation of lipid profile using semi-automated biochemistry analyzer (ERBA®). Random blood glucose will be estimated on the spot from a drop of blood at the time of sample collection, using a hand held glucometer (Accu-Chek®).

**Quality control.** Necessary measures as per the National guidelines for data quality in surveys [62] such as data collection by trained researchers, validation and pilot-testing of study instruments prior to use, meticulous checking and standardization of equipment, and reagents to be used, etc. will be taken to maintain data quality at the highest possible level.

**Data management and analysis.** All participants' data will be coded by assigning unique identifiers. These codes will be used to identify and link various parameters of the individual participants. Data from the filled questionnaires/clinical proforma as well as the laboratory analysis values will be entered into the latest version of Microsoft Excel spreadsheets. Statistician of the team will supervise data entry and to ensure error-free data at least 10% of the entered data will be randomly verified.

Once data entry is complete, hard copies of questionnaire/proforma will be stored in a secure archive in the institute. Similarly, all entered data will be stored in password protected computer systems with access restricted to the research team alone. The data collected/generated in this study will be stored for a period of ten years. The primary investigator will be in-charge of data safety and back-up maintenance. Anonymized data set will be shared publicly during publication of study results.

All statistical analyses will be performed using IBM SPSS Statistics (version 25). For ordinal and continuous variables, the values will be given as descriptive statistics (like mean, median, standard deviation, 95% confidence interval, interquartile range); and categorical variables will be given as numbers (percentages) and proportions. Association for different health aspects with pollution level of various parameters of air/water will be made using Pearson's chi-square test for independent categorical data. By considering the variability within and between groups; t-test and ANOVA will be used to analyze the mean differences between two and more groups respectively for various anthropological parameters, the concentration of air/water pollutants, etc. The Mann–Whitney U test and Wilcoxon signed ranks test will be used for independent and dependent for ordinal and continuous data with respect to different health aspects. Pearson and Spearman's correlation coefficient will be used to examine the association between different variables. Multiple linear and logistic regression models will be used to estimate the relationship for different neurological assessments with a set of the independent variable. For all statistical tests, two-sided P-values $<0.05$ will be considered to be significant.

**Ethics and dissemination.** Ethics approval has been obtained from the Institutional (Human) Ethics Committee, National Institute for Research in Environmental Health (No: NIREH/IEC-7-II/1027, dated January 7 2021). Before the initiation of research activities, local authorities of the selected village will be approached for consent and aid in logistics. Prior to the start of data collection written informed consent will be taken from each participant. Dissemination of the study findings will be done through submission of study technical report to the funding agency, distribution of health reports to all the participants while maintaining their confidentiality and finally, through publications.

## Discussion

This manuscript describes the protocol for a cross-sectional study being conducted to comprehensively assess the health effects of residing near viscose rayon and associated chemical manufacturing industries. Viscose rayon is one of the oldest man-made commercial fiber [63] that is still quite popular today with almost 4.5 million tons being manufactured annually

worldwide [64]. It is manufactured by chemically dissolving natural cellulose, commonly in the form of wood pulp [63], using a NaOH / $CS_2$ dissolution system [64]. However, during this process almost 25–30% of the $CS_2$ used is not recovered and escapes as emission into the ambient air [64]. In their review, Jiang et al., 2020 [64] report that for every ton of viscose produced, almost 20kg of exhaust and 300–600 tons of waste water containing heavy metals and other toxic residues are generated. Our study will produce up-to-date evidence specific to the Indian context which is especially relevant considering the fact that India is the second largest manufacturer of rayon, next only to China [63].

Chronic low dose exposure to $CS_2$, a major component of the viscose rayon industry exhaust, affects human physiology especially the cardiovascular and nervous system [10, 11, 65]. Studies have documented abnormalities in blood pressure [66], ECG findings [67], and lipid profile [68]. Similarly $CS_2$ exposure has been linked to peripheral neuropathy [69], abnormalities in colour vision [70] and neurobehavioral pattern [71]. Though fewer studies have been done to assess the effects of $CS_2$ on the respiratory system, Spyker et al., 1982 [72] showed alterations in pulmonary function. Effect on eyes, reproductive system and renal system have also been noted, although following prolonged heavy exposure [46]. Other gaseous pollutants emitted during viscose rayon and associated chemical manufacturing like $H_2S$, HCl, $Cl_2$, and $SO_2$ are known to affect human health, particularly the cardiorespiratory and nervous system [73–75].

Similarly, water and soil contamination occurring due to effluents being discharged by viscose rayon manufacturing industries has been reported in literature [76]. Effluents of textile industries using high amounts of salts may promote groundwater salinization due to the high mobility of salts/ions in the soil, and thus deteriorate drinking and irrigation water quality that may impact human health and land fertility/ crop productivity [77, 78]. The viscose fibre wastewater has also been shown to have a wide variety of residues ranging from heavy metals to cellulose and lignin remnants, most of which resist biodegradation [9, 79]. Previously published studies [80] report exposure of workers in viscose rayon industries to heavy metals like mercury, lead, etc. Chronic heavy metal exposure is again known to cause multiple adverse cardiorespiratory and neurological effects [81].

Most of the evidence regarding health effects of exposure to pollution emanating from viscose rayon and associated chemical manufacturing originates from outside India with there being hardly any published epidemiological evidence focussing on either workers in Indian industries [82] or nearby residents. However, the difficulties perceived by the residents living in Nagda industrial region has been scientifically documented [15] as well as has been highlighted in multiple other communication media [83]. Previous reports by local regulatory authorities [18] showed substantial heavy metal contamination of groundwater in and around Nagda region. Furthermore, the chemical industries in Nagda region are engaged in production of various chlorinated and sulphur-containing product (e.g., poly-aluminium chloride, bleaching powder, calcium chloride, chloromethane, chlorosulphonic acid) [84] which can potentially lead to introduction of other heavy metals like Aluminium and ions like $Cl^-$ and $SO_4^-$ ions into the surrounding groundwater. Therefore, this study is planned to collect pertinent and timely comprehensive information on the combined effect of the multiple toxic gases including $H_2S$, Sulphur oxides, HCl, $Cl_2$, etc. emanating as exhaust from the viscose rayon and associated chemical manufacturing industries as well as to assess the physiological parameters and heavy metal/ ion concentration in the groundwater in the study area.

Industrial pollution is a multidimensional issue since it occurs in multiple environmental matrices and in turn affects multiple organ systems in the human. Therefore, in our study we will focus on multiple objectives with an overarching goal to assess the potential health effects of exposure. A major strength of this study is therefore its multidisciplinary research team

with expertise of investigators ranging from health sciences like epidemiology and neurology to environmental sciences and basic science. To strengthen the association between pollutant concentration in environmental matrices and health effects, human exposure assessment will be done in a subset of the population to ascertain dose-response relation. Therefore, exposure to $CS_2$ will be confirmed using the urine azide test which has been reported to adequately reflect occupational $CS_2$ inhalation by published literature [34–36]. Furthermore, blood levels of heavy metals will also be determined to estimate exposure of villagers to toxic metals, viz. Pb, Al and Hg from various environmental matrices, viz. air, water and food/soil. However, pertaining to resource limitation, biomonitoring will be conducted in a subset of the total study population.

The current study also faces certain limitations. Long term follow-up of the participants to establish a temporal relationship between exposure and outcome is not possible in our cross-sectional study. Considering the ongoing dispute between local residents, viscose rayon manu-facturers and the local authorities, the current study prioritised timely delivery of baseline health effect information. Future research might therefore focus on longitudinal data collec-tion on the environmental health effects of viscose rayon manufacturing. Furthermore, past exposure assessment will be done through questionnaire data and is therefore, prone to recall bias. Although data will be collected for multiple relevant confounders including other envi-ronmental/meteorological, socioeconomic and diet/lifestyle factors, the risk of residual con-founding cannot be completely eliminated.

## Acknowledgments

We would like to acknowledge the technical and field staff of ICMR-NIREH, participants as well as the local authorities and health workers working in the villages where we conducted our exposure assessment and health outcome assessment surveys. We would also like to thank Dr. Vivek Parashar, Research Officer, R.D. Gardi Medical College Ujjain, Madhya Pradesh, India, for his support in creating the study area maps.

## Author Contributions

**Conceptualization:** Tanwi Trushna, Vikas Dhiman, Rajesh Ahirwar, Subroto Shambhu Nandi, Rajnarayan R. Tiwari.

**Data curation:** Dharma Raj.

**Funding acquisition:** Subroto Shambhu Nandi, Rajnarayan R. Tiwari.

**Investigation:** Tanwi Trushna, Satish Bhagwatrao Aher, Rajesh Ahirwar, Swasti Shubham, Subroto Shambhu Nandi.

**Methodology:** Tanwi Trushna, Vikas Dhiman, Satish Bhagwatrao Aher, Dharma Raj, Rajesh Ahirwar, Swasti Shubham, Subroto Shambhu Nandi.

**Project administration:** Subroto Shambhu Nandi, Rajnarayan R. Tiwari.

**Resources:** Subroto Shambhu Nandi, Rajnarayan R. Tiwari.

**Supervision:** Subroto Shambhu Nandi, Rajnarayan R. Tiwari.

**Visualization:** Tanwi Trushna, Dharma Raj.

**Writing – original draft:** Tanwi Trushna.

**Writing – review & editing:** Tanwi Trushna, Vikas Dhiman, Satish Bhagwatrao Aher, Dharma Raj, Rajesh Ahirwar, Swasti Shubham, Subroto Shambhu Nandi, Rajnarayan R. Tiwari.

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
