## [Decision Letter · Decision Letter 0]

25 Mar 2022

PONE-D-22-03418Environmental monitoring and health assessment in an industrial town in central India: A cross-sectional study protocolPLOS ONE

Dear Dr. Trushna,

Thank you for submitting your manuscript to PLOS ONE. After careful consideration, we feel that it has merit but does not fully meet PLOS ONE’s publication criteria as it currently stands. Therefore, we invite you to submit a revised version of the manuscript that addresses the points raised during the review process.

We look forward to receiving your revised manuscript.

Kind regards,

Mohan Lal Dotaniya, Ph.D.

Academic Editor

PLOS ONE

Journal Requirements:

"The study on which this protocol is based was funded as a consultancy project by the Aditya Birla Grasim Industry Ltd (Service No. 4700221679/104; dated 10.11.2020) [Funding issued to SN] as per directions of local pollution control authorities. The funders had no role in study design, data collection and analysis, decision to publish, or preparation of the manuscript"

We note that you have provided funding information that is not currently declared in your Funding Statement. However, funding information should not appear in the Funding section or other areas of your manuscript. We will only publish funding information present in the Funding Statement section of the online submission form. 

"I have read the journal's policy and the authors of this manuscript have the following competing interests:

The study on which this protocol is based was funded by the Aditya Birla Grasim Industry Ltd, which owns the major viscose rayon manufacturing unit in the study area (Birlagram Nagda) as per the directions of local pollution control authorities as a consultancy project. However, the funding agency has no role in design/conduct of the study, decision to publish, or preparation of the manuscript. No author received any direct payment from the industry with regards to their contribution to this manuscript or the study as a whole."

"I have read the journal's policy and the authors of this manuscript have the following competing interests:

The study on which this protocol is based was funded by the Aditya Birla Grasim Industry Ltd, which owns the major viscose rayon manufacturing unit in the study area (Birlagram Nagda) as per the directions of local pollution control authorities as a consultancy project. However, the funding agency has no role in design/conduct of the study, decision to publish, or preparation of the manuscript. No author received any direct payment from the industry with regards to their contribution to this manuscript or the study as a whole."  

We note that you received funding from a commercial source: Aditya Birla Grasim Industry Ltd

6. We note that you have referenced (iwari RR, Raghvan S, Tripathi S. Cardiological and neurological health effects in viscose rayon workers exposed to carbon disulphide. 2012. (Unpublished manuscript).) which has currently not yet been accepted for publication. Please remove this from your References and amend this to state in the body of your manuscript: (iwari RR, Raghvan S, Tripathi S. Cardiological and neurological health effects in viscose rayon workers exposed to carbon disulphide. 2012. (Unpublished manuscript). [Unpublished]”) as detailed online in our guide for authors

7. Your ethics statement should only appear in the Methods section of your manuscript. If your ethics statement is written in any section besides the Methods, please move it to the Methods section and delete it from any other section. Please ensure that your ethics statement is included in your manuscript, as the ethics statement entered into the online submission form will not be published alongside your manuscript. "

Additional Editor Comments:

MS was critically reviewed by the different reviewers. Please follow the journal guidelines before submission of the revised MS.

Reviewers' comments:

Reviewer's Responses to Questions

**Comments to the Author**

1. Does the manuscript provide a valid rationale for the proposed study, with clearly identified and justified research questions?

Reviewer #1: Yes

Reviewer #2: Yes

Reviewer #3: Yes

2. Is the protocol technically sound and planned in a manner that will lead to a meaningful outcome and allow testing the stated hypotheses?

Reviewer #1: Yes

Reviewer #2: Yes

Reviewer #3: Yes

3. Is the methodology feasible and described in sufficient detail to allow the work to be replicable?

Reviewer #1: Yes

Reviewer #2: Yes

Reviewer #3: Yes

4. Have the authors described where all data underlying the findings will be made available when the study is complete?

Reviewer #1: Yes

Reviewer #2: Yes

Reviewer #3: Yes

5. Is the manuscript presented in an intelligible fashion and written in standard English?

Reviewer #1: Yes

Reviewer #2: Yes

Reviewer #3: Yes

6. Review Comments to the Author

You may also provide optional suggestions and comments to authors that they might find helpful in planning their study.

Reviewer #1: The submitted manuscript is a protocol of a proposed environmental health study involving multiple

parameters. Though the traditional standard methods for analysis/assessment of various individual

parameters are already known, the comprehensiveness and wide scope of the study is a successful

novel attempt. The huge number of study participants, clinical examinations, biological assessment

(blood and urine), sampling and analysis designed for environmental matrices (air and water), lung

function test, ECG, neurological assessment proposed by the authors is a piece of wonderful work

and will serve as a ready reckoner for future environmental health studies.

Though the proposed protocol is for conducting environmental health study in and around man-made fibre and allied industrial settlement, the same can be used for diverse sectors with little

modifications considering the type and intensity of environmental contaminants.

Reviewer #2: Environmental pollution by viscose rayon textile industry is a matter of concern that has been neglected. The protocol will serve as an important document for the development of future studies in particular to environmental aspects of textile industry.

Following are the suggestions:

Spirometry is one of the important parameter of the study and hence detailed information should be provided in particular to regression equation used for calculation of predictive values. Also, details of interpretation of spirograms for determination of pulmonary impairments should be included in the manuscript.

Reviewer #3: The manuscript entitled “Environmental monitoring and health assessment in an industrial town in central India: A cross-sectional study protocol” submitted by Tanwi Trushna and co-workers is an industry specific study. However, the substitution/modification of parameters may serve as a standard protocol. The proposed study seems an interdisciplinary one which includes various areas such as environmental science, medical science, basic science, social science etc. The protocol is interesting and its comprehensiveness adds novelty. Thus, I recommend this manuscript for publication in PlosOne after necessary minor revision.

Minor revision requires for following aspects:

1. The list of abbreviation should be uniform.

2. The analytical methods used in water analysis should be described in details including pre-treatment before ICP-OES

determination.

3. Include the references for every standard analytical method.

7. PLOS authors have the option to publish the peer review history of their article (what does this mean?). If published, this will include your full peer review and any attached files.

Reviewer #1: No

Reviewer #2: No

Reviewer #3: No

---

## [Author Response · Author response to Decision Letter 0]

25 Apr 2022

Responses to reviewers

I. Points raised by the academic editor:

Comment 1: Please ensure that your manuscript meets PLOS ONE's style requirements, including those for file naming. 

Response: The manuscript has been revised to meet PLOS ONE's style requirements, including those for file naming

Comment 2: We note that the grant information you provided in the ‘Funding Information’ and ‘Financial Disclosure’ sections do not match. When you resubmit, please ensure that you provide the correct grant numbers for the awards you received for your study in the ‘Funding Information’ section.

Response: The required details have been provided in the cover letter.

Comment 3: Thank you for stating the following in the Funding Section of your manuscript: 

"The study on which this protocol is based was funded as a consultancy project by the Aditya Birla Grasim Industry Ltd (Service No. 4700221679/104; dated 10.11.2020) [Funding issued to SN] as per directions of local pollution control authorities. The funders had no role in study design, data collection and analysis, decision to publish, or preparation of the manuscript"

We note that you have provided funding information that is not currently declared in your Funding Statement. However, funding information should not appear in the Funding section or other areas of your manuscript. We will only publish funding information present in the Funding Statement section of the online submission form. 

"I have read the journal's policy and the authors of this manuscript have the following competing interests:

The study on which this protocol is based was funded by the Aditya Birla Grasim Industry Ltd, which owns the major viscose rayon manufacturing unit in the study area (Birlagram Nagda) as per the directions of local pollution control authorities as a consultancy project. However, the funding agency has no role in design/conduct of the study, decision to publish, or preparation of the manuscript. No author received any direct payment from the industry with regards to their contribution to this manuscript or the study as a whole."

Response: The funding information has been removed from the manuscript and the amended Funding Statement has been included within the cover letter.

Comment 4: Thank you for stating the following in the Competing Interests section: 

"I have read the journal's policy and the authors of this manuscript have the following competing interests:

The study on which this protocol is based was funded by the Aditya Birla Grasim Industry Ltd, which owns the major viscose rayon manufacturing unit in the study area (Birlagram Nagda) as per the directions of local pollution control authorities as a consultancy project. However, the funding agency has no role in design/conduct of the study, decision to publish, or preparation of the manuscript. No author received any direct payment from the industry with regards to their contribution to this manuscript or the study as a whole." 

We note that you received funding from a commercial source: Aditya Birla Grasim Industry Ltd

Response: The amended Competing Interests Statement explicitly stating the details of the commercial funder has been included within the cover letter.

Comment 5: In your Data Availability statement, you have not specified where the minimal data set underlying the results described in your manuscript can be found. PLOS defines a study's minimal data set as the underlying data used to reach the conclusions drawn in the manuscript and any additional data required to replicate the reported study findings in their entirety. All PLOS journals require that the minimal data set be made fully available. For more information about our data policy, please see http://journals.plos.org/plosone/s/data-availability.

Response: Since the current manuscript merely describes the protocol of a study and no results are yet generated or mentioned, we cannot as of now upload any data set. However, the data will be shared publicly during the publication of our findings after study completion. This point has been detailed in the cover letter and in the revised manuscript. [Line Number-375-376 of page Number-20 of the clean/unmarked version of the revised manuscript submitted] 

Comment 6: We note that you have referenced (iwari RR, Raghvan S, Tripathi S. Cardiological and neurological health effects in viscose rayon workers exposed to carbon disulphide. 2012. (Unpublished manuscript).) which has currently not yet been accepted for publication. Please remove this from your References and amend this to state in the body of your manuscript: (iwari RR, Raghvan S, Tripathi S. Cardiological and neurological health effects in viscose rayon workers exposed to carbon disulphide. 2012. (Unpublished manuscript). [Unpublished]”) as detailed online in our guide for authors

Response: Relevant modification has been done in Table 2 Line Number- 330 of page Number- 15-18 of the clean/unmarked version of the revised manuscript submitted.

Comment 7: Your ethics statement should only appear in the Methods section of your manuscript. If your ethics statement is written in any section besides the Methods, please move it to the Methods section and delete it from any other section. Please ensure that your ethics statement is included in your manuscript, as the ethics statement entered into the online submission form will not be published alongside your manuscript. "

Response: The detailed ethics statement has been written in the Methods section [Line Number- 391-399 of page Number- 21 of the clean/unmarked version of the revised manuscript submitted]. Ethics details have been removed from Acknowledgement section.

Comment 8: Please include captions for your Supporting Information files at the end of your manuscript, and update any in-text citations to match accordingly. Please see our Supporting Information guidelines for more information: http://journals.plos.org/plosone/s/supporting-information.

Response: No relevant supporting information needs to be submitted [The grant order and ethical approval documents had been previously uploaded as supporting information].

Comment 9: Please review your reference list to ensure that it is complete and correct. If you have cited papers that have been retracted, please include the rationale for doing so in the manuscript text, or remove these references and replace them with relevant current references. Any changes to the reference list should be mentioned in the rebuttal letter that accompanies your revised manuscript. If you need to cite a retracted article, indicate the article’s retracted status in the References list and also include a citation and full reference for the retraction notice.

Response: Modifications in the reference list [Line Number- 496-746 of page Number- 25-32 of the clean/unmarked version of the revised manuscript submitted]-

• In reference no. 11 and 67 of the previous list, a single article (Sieja et al., 2018) had been mistakenly cited as two separate citations- merged [Current list – reference no. 11].

• Reference no. 17 [Current list] added in Introduction (recent statistics of a previously cited pollution level)

• Reference no. 21 of the previous list that showed details about the province of Madhya Pradesh removed (link no longer active) and replaced with another existing reference [Current list – reference no. 19].

• Reference no. 39 of the previous list that contained details about hypertension and obesity definition from the National Centre for Disease Control updated to include functional weblink [Current list – reference no. 38].

• Reference no. 47 of the previous list that contained details about Carbon Disulphide from the WHO updated to include functional weblink [Current list – reference no. 46].

• Reference no. 83 of the previous list that contained details about Carbon Disulphide from the WHO updated to include functional weblink [Current list – reference no. 82].

II. Reviewer Comments

Reviewer#1

General comments on the manuscript: The submitted manuscript is a protocol of a proposed environmental health study involving multiple parameters. Though the traditional standard methods for analysis/assessment of various individual parameters are already known, the comprehensiveness and wide scope of the study is a successful novel attempt. The huge number of study participants, clinical examinations, biological assessment (blood and urine), sampling and analysis designed for environmental matrices (air and water), lung function test, ECG, neurological assessment proposed by the authors is a piece of wonderful work and will serve as a ready reckoner for future environmental health studies. Though the proposed protocol is for conducting environmental health study in and around man-made fibre and allied industrial settlement, the same can be used for diverse sectors with little modifications considering the type and intensity of environmental contaminants.

Response: Thank you for your recommendation.

Specific comments: 

Comment: Line 207: The sample processing/digestion techniques and equipment/instrument proposed to use is missing, include a brief description.

Response: As suggested, the sample processing and metal analysis part has been detailed along with mention of equipment and kits in the revised manuscript [Line Number-217-255 of page Number-10-12 of the clean/unmarked version of the revised manuscript submitted]. 

Comment: Line 226-227: Provide the reference for detection of chloride and sulphate using titration methods

Response: Estimation of chloride and sulphate levels in the groundwater will be carried out using ready-to-used kits (AQUACheck Sulphate and chloride testing kits, Himedia Laborities Pvt Ltd, India). The method and reference for the same have been provided in the revised manuscript. [Line Number-256-270 of page Number-12 of the clean/unmarked version of the revised manuscript submitted]

Comment: Line 233-234: Mention the approximate duration of storage

Response: The data will be stored for ten years as the approval for the same has been accorded by the institutional ethics committee. The details have been provided in the revised manuscript. [Line Number-373-374 of page Number-20 of the clean/unmarked version of the revised manuscript submitted]

Comment: Line 239: m?? Do you mean mL

Response: Yes; the typographical error has been corrected to mL and the section has been rewritten. [Line Number-217-255 of page Number-10-12 of the clean/unmarked version of the revised manuscript submitted]

Comment: Line 240: Provide the reference

Response: Appropriate reference has been provided in the revised manuscript. [Current Reference list Number 29-33 and 37]

Reviewer#2

Comment: Environmental pollution by viscose rayon textile industry is a matter of concern that has been neglected. The protocol will serve as an important document for the development of future studies in particular to environmental aspects of textile industry.

Following are the suggestions: 

Spirometry is one of the important parameter of the study and hence detailed information should be provided in particular to regression equation used for calculation of predictive values. Also, details of interpretation of spirograms for determination of pulmonary impairments should be included in the manuscript.

Response: Thank you for your recommendation. As suggested the details of interpretation of spirograms for determination of pulmonary impairments have been included in the manuscript [Line Number-337-339 of page Number-19 of the clean/unmarked version of the revised manuscript submitted].

Reviewer#3

Comment: The list of abbreviation should be uniform.

Response: Relevant modifications have been done all through the revised manuscript.

Comment: The analytical methods used in water analysis should be described in details including pre-treatment before ICP-OES determination

Response: As suggested by the esteemed reviewer, all analytical methods relating to water analysis have been detailed in the revised manuscript [Line Number-217-270 of page Number-10-12 of the clean/unmarked version of the revised manuscript submitted]. 

Comment: Include the references for every standard analytical method.

Response: References for all the standard analytical methods have been included (wherever previously missing) in the revised manuscript. For example for water analysis, standard analytical methods have been cited vide current reference list number 29-33 and 37.

---

## [Editor Report · Decision Letter 1]

31 May 2022

Environmental monitoring and health assessment in an industrial town in central India: A cross-sectional study protocol

PONE-D-22-03418R1

Dear Dr. Trushna,

We’re pleased to inform you that your manuscript has been judged scientifically suitable for publication and will be formally accepted for publication once it meets all outstanding technical requirements.

Kind regards,

Mohan Lal Dotaniya, Ph.D.

Academic Editor

PLOS ONE
---

## [Editor Report · Acceptance letter]

7 Jun 2022

PONE-D-22-03418R1 

Environmental monitoring and health assessment in an industrial town in central India: A cross-sectional study protocol 

Dear Dr. Trushna:

I'm pleased to inform you that your manuscript has been deemed suitable for publication in PLOS ONE. Congratulations! Your manuscript is now with our production department. 

Kind regards, 

on behalf of

Dr. Mohan Lal Dotaniya 

Academic Editor

PLOS ONE